# LEARNING OBJECT SEGMENTATION THROUGH A PARAMETRIC POLYGON REPRESENTATION

## ABSTRACT

Differentiable polygon (boundary-/contour-based) modeling for object instance segmentation remains an open problem in computer vision and deep learning. It also has been under-explored in the deep learning era, compared with its counterpart, bit-mask (region-based) modeling. In this paper, we present a method of differentiable polygon-based instance segmentation. As commonly done in the prior art, we assume a fixed topology, i.e., the number of vertices, $K$ is predefined and fixed (e.g., $K = 250$) in learning and inference. We address two modeling problems: i) The alignment between a predicted $K$-vertex polygon and a target ground-truth $L$-vertex polygon in learning, where $L$ varies significantly. We present PolygonAlign similar in spirit to RoIAlign used in bit-mask-based instance segmentation, which enables using a simple $\ell_2$ norm as the vertex prediction loss function in learning. ii) The parameterization of a $K$-vertex polygon. We present a variant of the active contour model, which consists of a learnable contour initialization module and an one-step vertex-aware refinement/updating module. The initialization is learned via an affine transformation decoupled vertex regression method. A polygon is parameterized by a translation vector, a rotation transformation matrix, and the vertex displacement vectors. In experiments, the proposed method is tested on the MS-COCO 2017 benchmark using the Sparse R-CNN framework. It obtains state-of-the-art performance compared with the prior art of polygon modeling methods. We also show the empirical upper-bound performance of the proposed method is much higher than all existing instance segmentation methods, which encourages further research on differentiable polygon modeling.

## 1 INTRODUCTION

Object instance segmentation aims to localize objects in images accurately in terms of their boundary. It remains a challenging problem in computer vision due to the large variations of object shape, pose, appearance and scale, and occlusions in images. A high-performing object instance segmentation system has a wide range of important applications such as autonomous driving, robot object grasping and manipulation and medical image analyses.

There are two modeling schema in object instance segmentation: bit-mask (region-based) modeling and polygon (boundary-/contour-base) modeling. Bit-masks and polygons are two dual representations of localizing objects in an image. The former is a pixel-wise dense representation, while the latter is a vectorized sparse representation. In fact, when annotating objects in images in collecting the training and testing data, polygons are used by human due to its simplicity in labeling. When it comes to modern machine deep learning systems, bit-mask modeling is the current dominant approach since it is straightforward to design differentiable loss functions to evaluate a predicted bit-mask with respect to a target ground-truth mask (e.g., the widely used pixel-wise cross entropy loss) at a predefined canonical resolution (e.g., $14 \times 14$). Tremendous progress have been made for bit-mask based object instance segmentation through the deep learning based systems (He et al., 2017; Kirillov et al., 2020; Liu et al., 2018; Chen et al., 2020; Fang et al., 2021), as shown in the leader board of MS-COCO object instance segmentation benchmark (Lin et al., 2014).

Polygon modeling lacks the simplicity of bit-mask modeling, and differentiable polygon modeling remains an open problem from the general shape analysis perspective. More specifically, **the challenge lies in:** *how to parameterize a polygon such that we can easily evaluate a predicted*

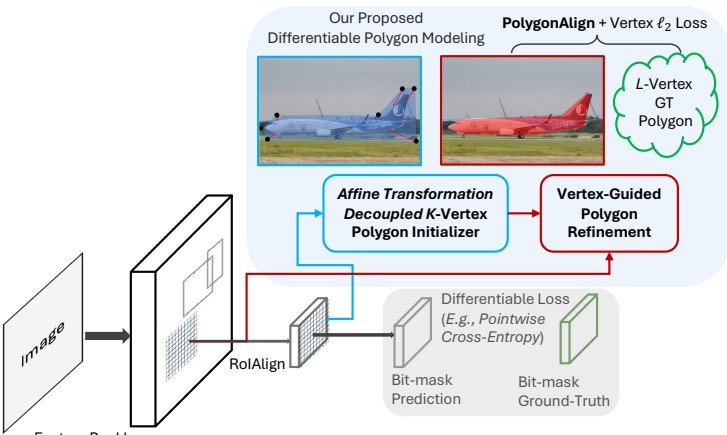

Figure 1: Illustration of the proposed differentiable polygon model for object instance segmentation in comparison with the dominant bit-mask scheme. Our proposed method consists of three components: (i) A novel PolygonAlign method that enables aligning predicted polygons of a fixed topology of a predefined $K$ vertices (e.g., $K = 250$) and annotated ground-truth $L$-vertex polygons of object instances, where $L$ can vary significantly from instance to instance, which in turn facilitates a simple vertex $\ell_2$ loss in end-to-end training. (ii) A learnable $K$-vertex polygon initializer from the region-of-interest (RoI) features of an object proposal (e.g., RoIAlign (He et al., 2017)). (iii) A one-step vertex-guided polygon refinement module.

*polygon with respect to a target ground-truth polygon, which in turn enables effective end-to-end learning to improve the polygon prediction based on the evaluated loss?*

In the literature, polygon modeling has been long explored in image segmentation since the seminal work of the active contour or snake model (Kass et al., 1988). Deep learning variants of active contour models (Gur et al., 2019; Zhang et al., 2022) have also been proposed in recent years and evaluated in challenging benchmarks such as the MS-COCO with promising results obtained. Those active contour models evolve some initial contours to fit objects' boundary, either using hand-crafted features and formulated under the energy minimization framework before the deep learning era or using deep learning features and formulated under some sort of recurrent/iterative neural network updating framework in more recent literature. Those methods often represent a polygon in the Cartesian coordinate system. Apart from this, there are works which use the polar coordinate system and often represent a polygon with a star-convex structure (i.e. radially convex), such as the PolarMask (Xie et al., 2021). Both lines of work assume fixed topology. Active contour based methods are sensitive to the initialization and the updating strategy. PolarMasks often can not represent complex polygons accurately due to its star-convexity constraint.

In this paper, we assume a fixed topology, i.e., the number of vertices, $K$ is predefined and fixed in learning and inference, as commonly done in the prior art. We use a sufficiently large number $K$ (e.g., $K = 250$). As illustrated in Fig. 1, we then address two problems in realizing differentiable polygon modeling for object instance segmentation,

*i) How to address the alignment or vertex correspondence problem between a always-$K$-vertex predicted polygon and a varying-$L$-vertex target ground-truth polygon, where $L$ often varies significantly from instance to instance?* As shown in Fig. 2, we re-represent annotated polygons by re-sampling $K$ vertices based on our proposed uniform Contour-Length-Fraction (CLF) sampling scheme. Intuitively, it is to "untie" a polygon to a line segment with the two end-points being the intersection point between the polygon and the $x$-axis, as shown in by the

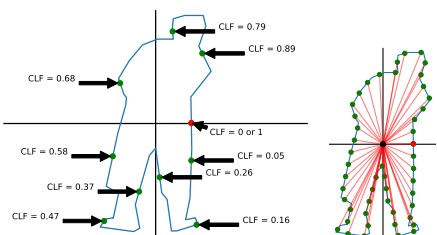

Figure 2: *Left:* Illustration of the proposed PolygonAlign via uniform contour-length-fraction (CLF) based vertex sampling. *Right:* An example of CLF based vertex sampling. See text for details.

end-points being the intersection point between the polygon and the $x$-axis, as shown in by the

red point. Then, we evenly divide the resulting line segment into $K$ pieces to sample the new vertices. The sampled $K$ vertices will be consistently sorted with respect to a predefined order (e.g., counter-clockwise), which enables building the vertex correspondence between a predicted polygon and a target ground-truth polygon in a straightforward way, i.e., **PolygonAlign**, similar in spirit to the RoIAlign He et al. (2017). It in turns facilitates using a simple $\ell_2$ norm as the vertex prediction loss function in learning to realize differentiable polygon modeling.

*ii) How to parameterize a $K$-vertex polygon regression module that can plug-and-play in existing object detection deep learning pipeline such as the Sparse R-CNN (Sun et al., 2021a)?* As shown in Fig. 3, we present a simple yet effective variant of the active contour model (Kass et al., 1988). We focus on developing a more expressive learnable polygon initializer, and on simplifying the iterative polygon updating to an one-step refinement. In our learnable polygon initializer, a polygon is parameterized by a translation 2D vector $\mathbb{T}_{1 \times 2}$, a rotation $2 \times 2$ matrix $\mathbb{R}_{2 \times 2}$, and $K$ vertex offset vectors $\mathbb{L}_{K \times 2}$. The rotation matrix induces the $K$ vertex displacement vectors to the global sorting order used in the vertex sampling. The translation vector can compensate for the position displacement error of the input detection bounding box, so the predicted polygon is capable of moving closer to the target polygon without being restricted by the detection bounding box. With the decoupled affine transformation, the regression of the $K$ vertex displacement vectors is locally calibrated, which facilitates faster learning convergence.

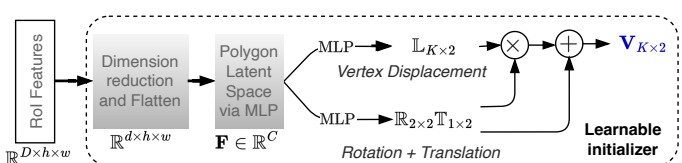

Figure 3: Illustration of the proposed polygon initializer using affine transformation decoupled vertex regression.

In experiments, the proposed method is tested on the challenging MS-COCO 2017 instance segmentation benchmark (Lin et al., 2014) using the Sparse R-CNN framework (Sun et al., 2021a). It obtains state-of-the-art performance compared with the prior art of polygon modeling methods. We also show the empirical upper-bound performance of the proposed method is much higher than all existing instance segmentation methods, which encourages further research on differentiable polygon modeling.

**Our Contributions.** This paper makes three main contributions to the field of polygon (boundary-/contour-based) modeling for instance segmentation: (i) It presents an intuitive PolygonAlign method that addresses the alignment between always-$K$-vertex predicted polygons (e.g., $K = 250$) and varying-$L$-vertex target ground-truth polygons, where $L$ varies significantly from instance to instance. The proposed PolygonAlign method facilitates using a single and simple mean squared error (MSE) function as the polygon prediction loss function in end-to-end learning. (ii) It presents an affine transformation decoupled vertex displacement based parameterization method for polygons to support the proposed PolygonAlign to maintain the predefined vertex correspondences without sacrificing modeling capability. It is used as the learnable polygon initializer under the active contour model. It also simplify the iterative updating with an one-step refiner. (iii) The proposed method obtains state-of-the-art performance in MS-COCO compared with the prior art of contour-based instance segmentation. It also analyzes the upper bound performance of the proposed method with some interesting observations.

## 2 METHOD

In this section, we present details of our PolygonAlign (Fig. 2) and our polygon parameterization method (Fig. 3). Then we present the integration between our polygon-based instance segmentation module and a state of the art object detection pipeline, the Sparse R-CNN method (Sun et al., 2021a).

### 2.1 POLYGONALIGN

We are motivated by the simplicity and expressivity of bit-mask (region-based) instance segmentation setting in which both predicted masks and target ground-truth masks are consistently kept at a predefined canonical resolution (e.g., a $14 \times 14$ grid based on RoIAlign in the Mask R-CNN (He et al.,

2017)) to facilitate simple pixel-wise cross-entropy loss functions used in end-to-end training, see Fig. 1. We aim to realize this in polygon-based instance segmentation. What would be the intuitive counterpart, **PolygonAlign**? While spatial resolution is the defining factor for aligning bit-masks / regions, the number of vertices is that for polygons. So, we focus on a fixed topology, i.e., the number of polygon vertices, denoted by $K$ (a hyperparameter in learning), is predefined and fixed in both training and inference. The number of vertices of predicted polygons can be easily controlled through the network design based on the hyperparameter setting. Then we are facing the problem of aligning always-$K$-vertex predicted polygons with varying-$L$-vertex target ground-truth polygons, where $L$ varies significantly from instance to instance? Our proposed PolygonAlign is simple yet effective with two steps as follows,

- We first re-sample vertices for ground-truth polygons from the labeled $L$ ones to the needed $K$ ones in training. As illustrated in Fig. 2, the proposed vertex re-sampling is a uniform sampling strategy based on the contour-length-fraction (CLF) to best preserve the geometry of polygons with the sampled discrete vertices. Our CLF mapping creates a map $S : \mathbb{R} \mapsto \mathbb{R}^2$. The map $S$ takes us from the space of contour-length-fractions, $[0, 1]$ onto $\mathbb{R}^2$ where the polygon vertices are defined. For example, this map can answer the question: *Given a fixed start point in the polygon, what is the endpoint that would form an arc that is $10\%$ of the length of the polygon?*

- After aligning the number of vertices between predicted polygons and ground-truth polygons, we need to maintain a consistent vertex-to-vertex order between them. Unlike the bit-mask modeling scheme in which the labels in the ground-truth mask are pixel-wise and dense, and the pixel-to-pixel alignment is already determined once the resolution is aligned. For two $K$-vertex polygons, the vertex-to-vertex assignment/correspondence is not fixed and subject to the design in learning. For simplicity, we keep a consistent order of the re-sampled $K$ vertices as illustrated in Fig. 2: The first and last vertices are the same, i.e., the intersection point between the polygon and the $x$-axis, and then a counter-clockwise order is adopted. With this setting, it is still non-trival to induce the same order for vertices of predicted polygons, e.g., based on their natural output entry order in the computation. To address this issue, we propose the affine transformation decoupled polygon parameterization method (elaborated below), which eliminates the need of introducing a certain sophisticated dynamic vertex matching component in learning in the prior art.

**Advantages of Our Proposed PolygonAlign.** Our PolygonAlign via CLF-based vertex sampling enables expressing complex polygons including those with concave and non-star-convex shapes, and even self-intersected ones, since we directly focus on polygon edges. Those types of polygons can not be accurately captured by star-convex structures used in PolarMask (Xie et al., 2020; 2021). Based on the vertex sampling and sorting, we resolve the vertex correspondence between predicted polygons and target ground-truth ones, which are consistent across all instances and throughout the learning, facilitating more stable optimization in training and better overall performance. This correspondence is often not utilized in the deep learning variants of active contour models. Instead, they usually need sophisticated designs in finding the correspondence for vertices of a predicted polygon on the fly as done in (Zhang et al., 2022). Unlike our PolygonAlign, DeepSnake's extreme point alignment scheme (Peng et al., 2020) creates an order that is highly discontinuous with respect to the contour of the polygon. Even small alterations to the contour can drastically change the alignment of the extreme points. The extreme point alignment method either down-samples or up-samples the polygon vertices without guaranteeing uniform distribution along the entire contour. For example, if the polygon already has the desired number of vertices, it will not be resampled, potentially leading to non-uniform spacing. If the ground truth polygon has too many vertices, vertices from the longest edges will just be removed. When adding new vertices, they are distributed evenly across their target edge, but this does not mean uniformity across the whole polygon. This also does not provide any instruction for alignment as our CLF does.

**PolygonAlign Enables Simple Vertex $\ell_2$ Loss.** With the proposed PolygonAlign, denote by $\mathbf{V}_{K \times 2}$ a predicted polygon, and by $\mathbf{V}^*_{K \times 2}$ the target re-sampled ground-truth polygon, with their vertex-to-vertex correspondences being the natural row indices. In learning, we use the simple mean squared error (i.e., $l$-2 norm) as the loss function,

$$\ell(\mathbf{V}_{K \times 2}, \mathbf{V}^*_{K \times 2}) = \frac{1}{K} \|\mathbf{V}_{K \times 2} - \mathbf{V}^*_{K \times 2}\|_2. \tag{1}$$

To sum up, the proposed PolygonAlign method enables a simple formulation for end-to-end polygon-based instance segmentation, playing a role similar in spirit to the RoIAlign method used in the Mask R-CNN for bit-mask based instance segmentation.

## 2.2 THE PROPOSED DIFFERENTIABLE POLYGON-BASED INSTANCE SEGMENTATION

The proposed method is a variant of the classic active contour or snake model (Kass et al., 1988) consisting of a learnable initializer and an one-step refiner, as illustrated in Fig. 3.

### 2.2.1 PARAMETERIZING POLYGONS VIA AFFINE TRANSFORMATION DECOUPLED VERTEX DISPLACEMENT REGRESSION

Without loss of generality, let $\mathbf{F}_C \in \mathbb{R}^C$ be the $C$-dim feature vector extracted from a feature backbone based on an object detection bounding box or an object detection center point. Our goal is to predict a $K$-vertex polygon (as the initialization) from $\mathbf{F}_C$ using a regression formulation,

$$\mathbf{V}_{K \times 2} = f(\mathbf{F}_C; \theta), \tag{2}$$

where $\theta$ collects the model parameters.

A straightforward method is to directly regress the vertex relative positions (i.e., offset vectors as shown by the center-vertex line segments in red in the right of Fig. 2) with respect to the position where $\mathbf{F}_C$ is computed. Due to the vertex correspondences assumption used in our PolygonAlign (Sec. 2.1), the direct regression method can not handle well the large variations of object pose, scale, and viewpoints, etc. in two-fold as follows.

**Affine Transformation Decoupled Vertex Offset Regression.** Consider a standing upright person and a laying-down person in an image, the PolygonAligh re-sampled and ordered vertices of their annotated polygons are actually not geometrically aligned. This "misalignment" is not known to the segmentation method which presumably uses the defined vertex correspondences. A simple solution is to allow the predicted vertices to learn to rotate to counter the "misalignment" on the fly. Furthermore, the anchor position of $\mathbf{F}_C$ can not be guaranteed to be sufficiently close to the true polygon center due to the fact that the object detection performance itself is bounded. This scenario is especially true in the early stages of the end-to-end training. So, we have the "misdisplacement" issue which is shared by all vertices and unknown to the segmentation method. A simple compensation is to allow the segmentation method to learn to re-place the anchor on the fly.

So, to address the above two issues, we present *an affine transformation decoupled vertex offset regression method*, as illustrated in Fig. 3. From the input $\mathbf{F}_C$, we learn,

$$\text{Rotation:} \quad \mathbb{R}_{2 \times 2} = f_{\mathbb{R}}(\mathbf{F}_C; \theta_{\mathbb{R}}), \tag{3}$$

$$\text{Translation:} \quad \mathbb{T}_{1 \times 2} = f_{\mathbb{T}}(\mathbf{F}_C; \theta_{\mathbb{T}}), \tag{4}$$

$$\text{Vertex Offset:} \quad \mathbb{L}_{K \times 2} = f_{\mathbb{L}}(\mathbf{F}_C; \theta_{\mathbb{L}}), \tag{5}$$

where $f_{\mathbb{R}}()$, $f_{\mathbb{T}}$ and $f_{\mathbb{L}}()$ are implemented using Multi-Layer Perceptrons (MLPs) for simplicity, and $\theta = (\theta_{\mathbb{R}}, \theta_{\mathbb{T}}, \theta_{\mathbb{L}})$ the model parameters of those MLPs. Then we predict the vertex positions (i.e., the initialization of the polygon) by,

$$\mathbb{V}_{K \times 2} = \mathbb{L}_{K \times 2} \cdot \mathbb{R}_{2 \times 2} + \mathbb{T}_{1 \times 2}, \tag{6}$$

where the translation vector is broadcasted to all vertices in the addition.

### 2.2.2 REFINING POLYGON PREDICTION VIA ONE-STEP VERTEX-GUIDED DEFORMATION

The above polygon initializer (Eqn. 6) focuses on all vertices, trying to do the best for predicting all of them at once by sharing the same input feature $\mathbf{F}_C$ (which itself is a holistic description, e.g., based on RoIAlign), the affine transformation, and the hidden layers in the vertex offset MLP. However, not all the vertices have the same difficulty in prediction. The initializer may over-shoot or under-shoot some vertices. A refinement module is entailed.

From the theory of the active contour model (Kass et al., 1988), we know that iterative updating / evolving with respect to some energy / loss functions plays an important role in finalizing the polygon prediction. On the other hand, we can learn from the success of bit-mask modeling in

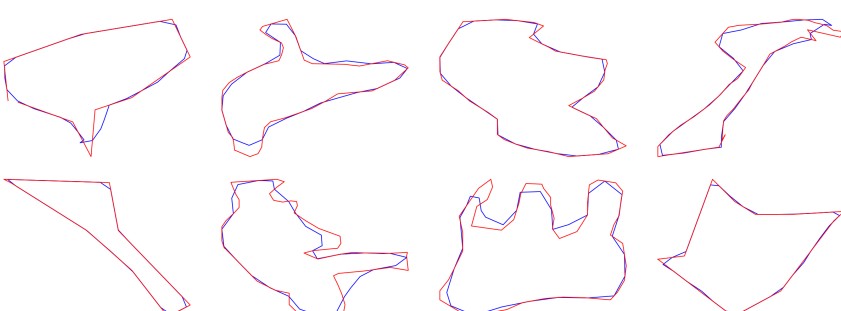

Figure 4: Examples of direct polygon fitting: ground truth polygons are shown in red and predicted polygons are in blue.

which pixel-wise (location sensitive) loss functions are used, and coarse mask guided point-based refinement has also found useful such as the PointRend method (Sitzmann et al., 2020). *The proposed polygon initializer can thus be improved in two aspects: using vertex-specific features and inducing interactions (or message passing) between vertices.* Our goal is to minimize the refining steps using one-step vertex-aware deformation to maintain the simplicity of the proposed polygon-based instance segmentation model.

Based on the initialized vertices $\mathbb{V}_{K\times 2}$, we extract features for each vertex from the feature backbone via the grid sample method, denoted by $\mathbb{F}_{K\times D}$. We then utilize the 1D circular convolution (Peng et al., 2020) to enforce interactions between vertices in learning the offsets. We have,

$$\Delta\mathbb{V}_{K\times 2} = f_{refine}(\mathbb{F}_{K\times D}; \theta_{refine}). \tag{7}$$

The final predicted polygon is computed by,

$$\mathbf{V}_{K\times 2} = \mathbb{V}_{K\times 2} + \Delta\mathbb{V}_{K\times 2}, \tag{8}$$

which is used in the loss computation (Eqn. 1).

## 3 EXPERIMENTS

In this section, we first design direct polygon fitting experiments to show the empirical upper bound performance of the proposed method, which sheds light on a few very interesting directions and encourages further research on differentiable polygon modeling. We then show results on the MS-COCO 2017 benchmark (Lin et al., 2014) and compare with the prior art of contour-based instance segmentation. Our method obtains state-of-the-art performance. We also show ablations studies on several aspects of the proposed method. **Our PyTorch source code will be publicly available.**

### 3.1 EMPIRICAL UPPER BOUND PERFORMANCE

We aim to study the empirical upper bound performance of our proposed polygon parameterization, i.e., the initializer itself. **We ask the question:** Is there a latent feature space of $\mathbf{F}_C$ (Eqns. 3,4,5) for our proposed polygon model to reach very high performance, and how high could it be? Our direct fitting experiments show that the proposed polygon parameterization method can reach very high performance, indicating its underlying effectiveness.

**Experiment I: Direct Polygon Fitting by Jointly Optimizing the Input $\mathbf{F}_C$ and the Polygon Model.** We use a set of 5000 polygons randomly sampled from the MS-COCO `train` set. We jointly train the input feature vectors representing each polygon $\mathbf{F}_C \in \mathbb{R}^{5000\times C}$ ($C = 256$, as model parameters) and the polygon initializer for 300 iterations using the full batch based optimization. We performed the AP evaluation on the same set of 5000 polygons used in training since this experiment involves optimizing a sample-specific polygon query directly.

For the models with the number of vertices $K = 50, 120, 250$, their APs are: $83.2\%, 82.5\%$ and $81.9\%$, respectively. These results clearly show the learnability and modeling capability of the proposed polygon parameterization. Fig. 4 shows some examples.

**Experiment II: Fitting Polygons in an Alternative Polygon Latent Feature Space.** In Experiment I, the latent feature space $\mathbf{F}_C$ is directly optimized without any explicit constraints, which may be too difficult to be transported from the raw image space. In this experiment, we define an alternative polygon latent space that is learned from input bit-masks of polygons using an encoder network. We jointly train the encoder network and the polygon model. We use the same set of 5000 polygons as in Experiment I. We use the mini-batch size of 32 and train 300 epochs.

To evaluate this experiment, we used a validation set of 5000 polygons, all randomly selected from the MS-COCO `val` set and unseen during training. For the models with the number of vertices $K = 50, 120, 250$, their APs are: $81.9\%, 82.7\%$ and $83.8\%$, respectively.

Although the inputs to this experiment are ground-truth bit-mask representation of a shape, converting them to a polygon representation remains a nontrivial task, this experiment verifies that there is a constrained polygon latent feature space, alternative to the directly optimized one, which also supports high-performing polygon based instance segmentation using our parameterization.

## 3.2 Instance Segmentation in MS-COCO

**Data and Metrics.** The MS-COCO instance segmentation benchmark (Lin et al., 2014) is one of the challenging and large-scale datasets aiming for instance segmentation in the wild. It contains 115k training, 5k validation, and 20k testing images with 80 object categories. We use the MS-COCO provided evaluation protocol in the evaluation. We train our models end-to-end on the `train` set. We compare with the prior art on the `test-dev 2017` set. We do ablation studies using the `val 2017` set.

**Settings.** We choose the Sparse R-CNN pipeline in our experiments, which is one state-of-the-art fully end-to-end object detection pipelines. Following the DETR framework (Carion et al., 2020), Sparse R-CNN (Sun et al., 2021b) also exploits a query-based design for end-to-end object detection with a set prediction formulation. We use implement our model using the mmdetection (Chen et al., 2019) PyTorch package which provides an off-the-shelf implementation for the Sparse R-CNN (Sun et al., 2021a).

For the polygon initializer (Fig. 3): $D = 256, h = w = 14$, so the RoI features are in $\mathbb{R}^{256 \times 14 \times 14}$ (see Fig. 1). We set $C = 64$, so the input feature vector $\mathbf{F}_C \in \mathbb{R}^{64}$.

- The polygon initializer first reduces the RoI feature dimension from 256 to 8 and then flattens the RoI, resulting in features in $\mathbb{R}^{2048}$ ($2048 = 14 \times 14 \times 8$).

- It then applies a MLP consisting of 4 hidden layers of dimensions $(1024, 1024, 512, 512)$ using the ELU activation function, and the output layer of dimension $C = 64$. The vertex offset MLP (Eqn. 5) consists of 3 hidden layers of dimensions $(2C, 3C, 4C)$ using the ELU activtion function, and the output layer of dimension $(K - 1) * 2$ (where $K$ is the number of vertices).

- The affine transformation MLP (Eqns. 3, 4) consists of 3 hidden layers of dimensions $(256, 256, 256)$ using the ELU activtion function, and the output layer of dimension 6. For the refiner (Eqn. 7), the input $\mathbb{F}_{K \times D}$ is extracted from the feature backbone using grid sample based on the predicted initial polygon. The refiner consists of 4 layers of circular 1D convolution using the kernel size 3 and the GELU activation function, with dimensions $(512, 512, 256, 256)$, and the output layer computes the updated vertex offset.

We use two backbones, ResNet-50 and ResNet-101 (He et al., 2016), both of them are pretrained on ImageNet (Deng et al., 2009). We use 100 queries in the Sparse R-CNN and its vanilla object detection average precision (AP) on MS-COCO `val` set is 37.9 in the mmdetection, which is improved after the integration of our polygon model (see Sec. 3.3). We utilize the AdamW optimizer with the initial learning rate 0.001 and weight decay 0.001. The learning rate for the pretrained feature backbones is decreased by a multiplier 0.1. We use the basic data augmentation (resizing to (1333, 800) by keeping the aspect ratio and random left-right flipping). We train our model with both 12 epochs (i.e., the 1x schedule) and 24 epochs (i.e., the 2x schedule).

**Results.** Table 1 shows the performance comparison on the MS-COCO `test-dev` set. Our method obtains the best instance segmentation performance compared to the prior art of contour-based modeling. With the same feature backbone and training epochs, our method outperforms PolarMask++ (Xie et al., 2021) by 1.4%. Compared with E2EC (Zhang et al., 2022) which utilizes a

Table 1: Performance comparison with the prior art of contour-based instance segmentation on the MS-COCO `test-dev` set. "+MS" in PolySnake represents the multi-scale contour refinement module.

| Method | Venue | Backbone | Epochs | AP | $AP_{50}$ | $AP_{75}$ |
|---|---|---|---|---|---|---|
| PolarMask (Xie et al., 2020) | *CVPR'20* | Res-101 | 24 | 32.1 | 53.7 | 33.1 |
| PolarMask++ (Xie et al., 2021) | *TPAMI'21* | Res-101 | 24 | 33.8 | 57.5 | 34.6 |
| DeepSnake (Peng et al., 2020) | *CVPR'20* | DLA-34 | 160 | 30.3 | - | - |
| E2EC (Zhang et al., 2022) | *CVPR'22* | DLA34 | 140 | 33.8 | 52.9 | 35.9 |
| PolySnake (Feng et al., 2023) / +MS | *arXiv'23* | DLA34 | 250 | 34.5 / 34.9 | - | - |
| Ours | - | Res-50 | 12 | 32.1 | 54.2 | 33.1 |
| | | Res-50 | 24 | 33.3 | 55.7 | 34.4 |
| | | Res-101 | 12 | 33.2 | 55.9 | 34.3 |
| | | Res-101 | 24 | **35.2** | **58.4** | **36.7** |

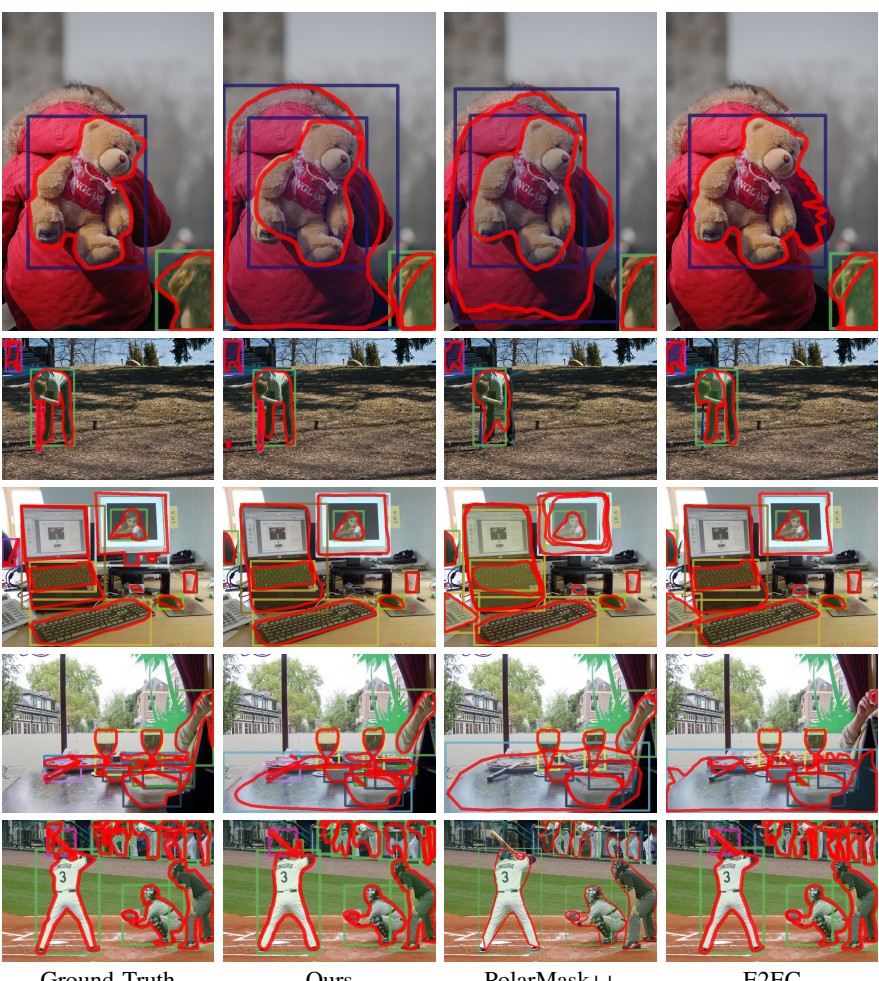

Ground-Truth      Ours      PolarMask++      E2EC

Figure 5: Qualitative comparison of contour-based instance segmentation on the MS-COCO *val* set between our method, PolarMask++ (Xie et al., 2021) and E2EC (Zhang et al., 2022). The results of PolarMask++ and E2EC are visualized using their released model checkpoints and codes. Compared to E2EC in the 1st row, our method shows smoother boundaries. In the 2nd row, the legs of the baseball player segmented by our method exhibit better refinement and representation compared to PolarMask++.

learnable initializer and a more sophisticated global and local deformation updating strategy, and is trained with much longer epochs (140 vs 24), our method increases the AP by 1.4% too. Compared with a most recent preprint work on arXiv, PolySnake (Feng et al., 2023) which uses an even more

sophisticated design of the updating (which may lead to the even longer training epochs, 250 vs 24), including a multi-scale refinement module, our method also obtains better performance. Fig. 5 shows qualitative comparison between different method. Overall, our method shows more faithful polygon predictions compared with PolarMask++ (e.g., the person in the 2nd row), and smoother polygon predictions than E2EC2 (e.g., the Teddy bear in the 1st row).

## 3.3 ABLATION STUDIES

We conduct two ablation studies on the MS-COO `val` set using the Res-50 backbone and the 12-epoch schedule.

**The effect of learning the affine transformation in our polygon parameterization.** As discussed in Sec. 2.2.1, we propose to decouple the affine transformation from the vertex displacement regression to help the network to better cooperate with our proposed PolygonAlign resampling. Table 2 shows the comparison. The affine transformation shows positive effects albeit not very significant.

Table 2: Ablation study on the affine transformation.

| #Vert., $K$ | Affine Trans. | $AP_{Det}$ | AP | $AP_{50}$ | $AP_{75}$ |
|---|---|---|---|---|---|
| 250 | ✗ | 40.5 | 31.5 | 53.3 | 32.2 |
| 250 | ✓ | **40.6** | **31.8** | **53.6** | **32.7** |

**The number of vertices in PolygonAlign.** We keep the number of vertices sufficiently large ($K = 250$) in our main experiments to ensure the expressivity based on our intuitive thoughts.

Table 3 shows the comparison. Although the model with $K = 250$ obtains the best instance segmentation performance, we see that the model with $K = 50$ is better than the model with $K = 120$. We first hypothesize that it

Table 3: Ablation study on the number of vertices.

| #Vert., $K$ | Aux. | Affine Trans. | $AP_{Det}$ | AP | $AP_{50}$ | $AP_{75}$ | Vert. Resampling Quality |
|---|---|---|---|---|---|---|---|
| 50 | ✓ | ✓ | **40.7** | 31.6 | 53.5 | 32.5 | 96.50 |
| 120 | ✓ | ✓ | 40.3 | 31.4 | 52.9 | 32.2 | 96.81 |
| 250 | ✓ | ✓ | 40.6 | **31.8** | **53.6** | **32.7** | 96.82 |

may be caused by a caveat in our uniform CLF-based vertex re-sampling (Fig. 2): Its uniformity can not guarantee to cover all the original annotated vertices very faithfully in the re-sampling when the number of re-sampling vertices is not sufficient large. But it turns out that this not the case. We compute the AP between the re-sampled polygons and the original annotated polygons as the vertex re-sampling quality. The last column in Table 3 shows that there is no obvious differences. Then it might be related to the difficulty of the optimization, e.g., a 120-vertex polygon predictor might just be unlucky and get stuck in the wrong optimization path purely due to the change in optimization landscape between different vertices. We note that this also could be simply caused by the common performance variations due to different training noises since we compare them using just one round of experiments. We leave a more comprehensive ablation study for the future work.

## 3.4 LIMITATIONS

One main limitation lies in the performance gap between the empirical upper bound and the performance evaluated in the MS-COCO test benchmark, which means there are a lot room for improvement in terms of integration designs and optimization strategies. Both experiments in Sec. 3.1 indicate that pushing the performance boundary of polygon / contour based instance segmentation entails careful rethinking on the design of feature backbones which have been mostly studied under the hood of bit-mask based modeling. One potential solution is to leverage powerful pretrained backbones such as the Segment Anything Model (SAM) (Kirillov et al., 2023) together with parameter-efficient fine-tuning methods such as Low-Rank Adaptation (LoRA) (Hu et al., 2022).

## 4 RELATED WORK

In this section, we briefly summarize the related work on instance segmentation (please see (Sharma et al., 2022) and (Minaee et al., 2021) for recent comprehensive surveys).

**Bit-Mask Modeling for Instance Segmentation.** The bit-mask (region-based) modeling scheme is the current state-of-the-art method in object instance segmentation, with tremendous progress achieved. Popular methods include two-stage detect-then-segment pipelines such as the Mask R-CNN (He et al., 2017), the Path Aggregation Networks (PANets) (Liu et al., 2018), the PointRend (Kir-

illov et al., 2020), and the more recent query-based designs, e.g., the Query2Instance (Fang et al., 2021), and single-stage anchor-free mask-prototype-assembling pipelines such as the YOLACT (Bolya et al., 2019) and the BlendMask (Chen et al., 2020), to just name a few. Our motivation in this paper is how to exploit the simplicity of designing differential loss functions with bit-mask based models into polygon-based modeling pipelines. To that end, we realize two aspects: i) The alignment are straightforward between predicted bit-masks and target ground-truth ones, which enables using dense pixel-wise differentiable loss functions. The alignment needs to be carefully handled between predicted and target polygons. ii) The predicted bit-masks are of the same 2D spatial structures as the ground-truth ones. The predicted polygon is essentially a chain of vertices without an embedded order of vertices that are aware of geometric variations of ground-truth polygons. We address the two aspects by proposing simple solutions.

**Contour Modeling for Instance Segmentation.** In general, comparing two polygons of different topologies remains an open problem. For example, to calculate the typical metric to gauge how similar two polygons $A$ and $B$ are using the Intersection-over-Union (IoU), we would first have to generate two new polygons from them, the union $A \cup B$ and the intersection $A \cap B$. The calculation of $A \cup B$ and $A \cap B$ itself is non-trivial. To calculate $A \cup B$ one would have to resort to "polygon clipping" algorithms such as the Sutherland-Hodgman algorithm (Sutherland & Hodgman, 1974). Since these algorithms are typically non-differentiable or inefficient for the purposes of training a model, a surrogate loss has to be created to be able to create an efficient differentiable metric in traning. To this end, methods such as differentiable rendering (Kato et al., 2018; Loper & Black, 2014) are used to create a pixel-wise mask from a polygon in a differentiable manner, once a pixel-wise mask is created a typical loss such as the DICE loss (Pan et al., 2019) or binary cross-entropy can be used and backpropagated to the polygon parameters through the differentiable rendering step as done in (Gur et al., 2019) which integrates the Active Contour or Snake Model (Kass et al., 1988) and differentiable rendering. Those methods have mainly been studied for building segmentation and in medical imaging, where polygons to be segmented are of relatively simple structures.

To eliminate the need of differentiable rendering. Other work focus on developing loss functions that do not rely on polygon-to-pixel-wise-mask conversions by designing alternative parameterization for polygons. In PolarMasks (Xie et al., 2020; 2021) and LSNets (Duan et al., 2021) a polygon is re-represented as polar coordinates at discrete and fixed sampling angles using a star-convex structure. In implementation, they use center-offset based regression, directly regressing the positions of vertices based on the features extracted at the center. To further leverage vertex-specific features for better vertex position regression, different variants of the active contour model (Kass et al., 1988) have been proposed (Ling et al., 2019; Peng et al., 2020; Wei et al., 2020; Liu et al., 2021; Zhang et al., 2022; Feng et al., 2023) with different methods used for contour initialization and iterative updating. The contour initialization are heuristic in (Ling et al., 2019; Peng et al., 2020; Wei et al., 2020; Liu et al., 2021), and learnable in (Zhang et al., 2022; Feng et al., 2023). In the iterative updating, predefined and fixed vertex correspondences between predicted polygons and target ground-truth ones are used in (Ling et al., 2019; Peng et al., 2020; Wei et al., 2020; Liu et al., 2021). Dynamic matching based on the Douglas-Peucker algorithm (Douglas & Peucker, 1973) is used in (Zhang et al., 2022). In (Feng et al., 2023), multi-scale refinement is used with multi-stage training and many different loss terms in a sophisticated design. Compared with the prior art, both the proposed PolygonAlign method and the affine transformation decoupled polygon parameterization are novel, and much simpler.

## 5 CONCLUSION

This paper proposes a method of differentiable polygon modeling for object instance segmentation under the active contour / snake modeling framework. It addresses two modeling problems. It presents the PolygonAlign that utilizes a contour-length-fraction (CLF) based vertex re-sampling strategy for aligning always-$K$-vertex predicted polygons and varying-$L$-vertex target ground-truth polygon using a simple $l$-2 norm in learning. It also presents the affine transformation decoupled vertex displacement regression method for polygon parameterization that cooperates with the PolygonAlign. The proposed method is tested in MS-COCO instance segmentation benchmark with state-of-the-art performance obtained compared with the prior art of contour-based instance segmentation. Different aspects of the proposed method are analyzed. The empirical upper bound performance of the proposed method is investigated using to direct polygon fitting experiments, from which some interesting observations are made, encouraging further research on differentiable polygon modeling.

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
