# OpenReview forum: "Differentiable Polygon Modeling for Object Instance Segmentation"
_ICLR.cc/2025/Conference — ICLR 2025 Conference Withdrawn Submission_

### Official Review · Reviewer_ennP · 2024-11-03

**Soundness:** 2
**Presentation:** 2
**Contribution:** 2
**Rating:** 5
**Confidence:** 3

**Summary:**

The paper focuses on polygon-modeling-based instance segmentation. It proposes PolygonAlign to address the alignment between always-K-vertex predicted polygons (e.g., K = 250) and varying-L-vertex ground-truth polygons. It presents an affine transformation decoupled vertex displacement based parameterization method for polygons to support the proposed PolygonAlign.

**Strengths:**

1. The paper noted how to parameterize a polygon when the number of convex is different between predicted polygon and ground-truth polygon. The method is somewhat technically sound.

2. The paper attempts to implement the method in end-to-end object detector.

3. The experimental settings is presented in a detailed way, which benefits for reproducing.

**Weaknesses:**

1. It is unclear that what is the superiority of the polygon-modeling-based instance segmentation over the bit-mask modeling method. The performance of the polygon-modeling-based instance segmentation is also inferior to the bit-mask modeling method, e.g., SOLO [r1], SOLOv2 [r2], CondInst [r3]. This makes the significance of this work unclear.

2. The paper did not report the runtime latency. So the efficiency of the proposed method is unclear.

3. It is evidenced by Table 3 that a small $K$ is a better choice. Why the paper spend so many effort in presenting the method with a large $K=250$?

4. It would be better present the results of Sec. 3.1 at tables.

[r1] Wang X, Kong T, Shen C, et al. SOLO: Segmenting objects by locations. ECCV 2020.

[r2] Wang X, Zhang R, Kong T, et al. SOLOv2: Dynamic and fast instance segmentation. NeurIPS 2020.

[r3] Tian Z, Shen C, Chen H. Conditional convolutions for instance segmentation. ECCV 2020.

**Questions:**

see weaknesses

---

### Official Review · Reviewer_5PMT · 2024-11-03

**Soundness:** 3
**Presentation:** 3
**Contribution:** 2
**Rating:** 5
**Confidence:** 4

**Summary:**

This paper proposes a simple yet effective contour-based instance segmentation model. The model is built on Sparse R-CNN detector and addresses two modeling problems: (1) the alignment between the predicted and the labeled vertices; (2) The parameterization of a K-vertex polygon. The experiment results show the performance advantage over pervious contour-based instance segmentation methods.

**Strengths:**

-	The paper is well-written and easy to follow and its motivation is clear.
-	The proposed method is simple yet effective, without the long training schedule and repeated refinements, yet exhibits significant performance advantages over previous methods.

**Weaknesses:**

-	The effectiveness of the proposed method is not clear enough. Tables 2 and 3 show that the proposed modifications (i.e., larger K, affine transformations) both have a minor impact on the results. So what led to the performance improvement over previous work? Is it the proposed CLF sampling strategy or simply because it is based on a better detector (i.e., Sparse R-CNN)?
-	Besides, ablation experiments on CLF-based vertex re-sampling are lacking.
-	Fig. 5 does not show a clear advantage and instead, the contours of the proposed method appear to be over-smooth.
-	Lack of inference speed comparison between the proposed method and previous methods.
-	There are a lot of typos, for example, PolygonAligh (L240), use (L349), MS-COO (L440), repeated citation (L625 and L629).

**Questions:**

-	Not all the instance segmented datasets are labeled with polygon, e.g. Cityscapes dataset is labeled with mask, is the proposed method applicable for this type of annotation?
-	How does the proposed method handle objects with more complex topologies, such as donuts?
-	In L322, why more K results to lower AP? Shouldn't a larger K lead to finer polygons?

---

### Official Review · Reviewer_EvJ3 · 2024-11-04

**Soundness:** 2
**Presentation:** 2
**Contribution:** 2
**Rating:** 5
**Confidence:** 4

**Summary:**

This paper discusses the polygon-based instance segmentation problem. To alleviate the difficulty of contour representation and learning, the authors proposed an approach to align the prediction and the ground truth of variant vertex numbers, an affine transformation-based parameterization method, and a refiner for precise regression. The authors conducted experiments on the MS-COCO instance segmentation benchmark and provided some experimental analysis.

**Strengths:**

1. The paper is well-written and easy to follow. The key challenges of polygon-based instance segmentation are described clearly. The proposed method for polygon alignment and coarse-to-fine refinement is reasonable.
2. The authors provided the method's performance with different training epochs, which could help set up a fair comparison with related works.
3. The authors provided some ablation studies to show the influence of different numbers of polygon vertices.
4. There are pilot experiments to study the upper bound of the method, which is very useful to help reveal the representation method's ability.

**Weaknesses:**

1. The advantage of using polygons instead of bit masks as the prediction target is unclear. It is well-known that polygons are harder to learn and are limited to only connected instances. The authors did not discuss what taking polygons as training targets brings to us. Therefore, from this view, the paper is not well motivated.
2. The performance is moderate. The results in Table 1 do not show the instance segmentation results of the Sparse R-CNN method, which is the base model used by the authors and should be an important baseline. Since previous approaches listed in Table 1 applied different base models, the comparison cannot prove the method's advantage convincingly. Besides, Table 2's results also show that the improvement of the affine transformation is marginal.
3. The original Sparse R-CNN reports AP 42.8%~46.4% on the COCO val set with different backbones. However, the authors claimed it can only achieve 37.9% (line 369). Therefore, the reviewer suggests aligning the base model with the original Sparse R-CNN for a more convenient comparison.
4. It lacks ablation studies of the refinement strategy.

**Questions:**

Please refer to the weaknesses for details. The major concerns are about motivation and performance.

---

### Official Review · Reviewer_5xmN · 2024-11-04

**Soundness:** 2
**Presentation:** 3
**Contribution:** 2
**Rating:** 3
**Confidence:** 3

**Summary:**

The paper introduces a novel framework for object instance segmentation based on differentiable polygon modeling, addressing instance segmentation via a polygon-based approach in contrast to widely used bit-mask methods.
It proposes PolygonAlign, which utilizes a contour-length-fraction (CLF) based vertex re-sampling strategy for aligning fixed K-vertex predicted polygons with varying L-vertex target ground-truth polygons using a simple L2 norm in learning.
It also presents an affine transformation decoupled vertex displacement regression method for polygon parameterization that cooperates with PolygonAlign.
It achieves SOTA results on MS-COCO among contour-based instance segmentation methods. The work addresses important challenges in making polygon-based approaches differentiable and competitive with mask-based methods.

**Strengths:**

- The paper is generally well-written and clearly structured. The figures effectively illustrate the key concepts.
- It is well-motivated and properly contextualized within related works.

**Weaknesses:**

- The performance improvements seem marginal. For a paper in 2024, 35 AP on COCO is not satisfactory. To showcase the method's effectiveness, the authors should compare with stronger baselines using better backbones/training protocols. Otherwise, it is not convincing why the method is more promising than other approaches.
- The computational complexity and inference time analysis is missing.
- The method still relies on fixed topology (K vertices), limiting its flexibility.
- The authors should verify their method on more datasets such as Cityscapes, LVIS, etc.

Overall, the performance of this paper is not strong enough for this competitive instance segmentation task. If the authors believe polygon-based methods are promising, they should revise their experimental results with better baselines or find a better experimental setting (such as for improving human annotation).

**Questions:**

Please see the weaknesses part.

---

### Note · Authors · 2024-11-25

**Comment:**

We thank the reviewers for their valuable time and efforts.

**Withdrawal Confirmation:**

I have read and agree with the venue's withdrawal policy on behalf of myself and my co-authors.